# Transforming Pedagogical Practices and Teacher Identity Through Multimodal (Inter)action Analysis: A Case Study of Novice EFL Teachers in China

**DOI:** 10.3390/bs15081050

**Published:** 2025-08-03

**Authors:** Jing Zhou, Chengfei Li, Yan Cheng

**Affiliations:** 1School of Foreign Languages, Chaohu University, Chaohu 238000, China; 2School of Communication Studies, Auckland University of Technology, Auckland 1010, New Zealand; 3School of Foreign Languages, Huazhong University of Science and Technology, Wuhan 430074, China; cflee@hust.edu.cn; 4School of Foreign Languages, Zhejiang Gongshang University, Hangzhou 314423, China; 23010130001@pop.zjgsu.edu.cn

**Keywords:** multimodal (inter)action analysis, novice EFL teachers, teacher identity development, multimodal pedagogical strategies, China

## Abstract

This study investigates the evolving pedagogical strategies and professional identity development of two novice college English teachers in China through a semester-long classroom-based inquiry. Drawing on Norris’s Multimodal (Inter)action Analysis (MIA), it analyzes 270 min of video-recorded lessons across three instructional stages, supported by visual transcripts and pitch-intensity spectrograms. The analysis reveals each teacher’s transformation from textbook-reliant instruction to student-centered pedagogy, facilitated by multimodal strategies such as gaze, vocal pitch, gesture, and head movement. These shifts unfold across the following three evolving identity configurations: compliance, experimentation, and dialogic enactment. Rather than following a linear path, identity development is shown as a negotiated process shaped by institutional demands and classroom interactional realities. By foregrounding the multimodal enactment of self in a non-Western educational context, this study offers insights into how novice EFL teachers navigate tensions between traditional discourse norms and reform-driven pedagogical expectations, contributing to broader understandings of identity formation in global higher education.

## 1. Introduction

Within the contemporary educational reforms in Chinese higher education, College English (CE) teachers teach non-English majors at tertiary institutions in China by fostering language proficiency and intercultural competence ([82]). Over the past decade, a surge in hiring new CE teachers has accompanied the compulsory status of College English courses across Chinese universities ([166]). However, the transition from novice to professional educator is often challenging, particularly in terms of identity formation, pedagogical knowledge consumption, and teaching efficacy improvement ([157]; [158]).

Traditionally, English language instruction in Chinese higher education has historically been characterized by teacher-centered pedagogies, emphasizing rote memorization and standardized test performance, particularly due to the influence of high-stakes assessments such as College English Tests (CET-4 and CET-6) ([47]). In such settings, teachers are positioned as authoritative knowledge providers, and students are positioned as recipients. This model has faced criticism for limiting students’ communicative competence ([114]), stifling innovative thinking ([44]), reducing learner autonomy ([29]), and hindering the development of higher-order thinking skills ([125]). Classrooms often follow the traditional Initiation–Response–Feedback (IRF) pattern, limiting interactive engagement and student autonomy ([3]).

In response, recent reforms highlight the importance of fostering autonomous learning, critical thinking, and intercultural awareness as essential competencies for modern education ([95]). For novice EFL teachers, this shift presents not only methodological challenges but also requires rethinking their professional identities. As [64] ([64]) argue, foreign language educators must develop symbolic competence, which is the ability to convey and interpret meaning through verbal and non-verbal modes such as gesture, gaze, and spatial positioning. Teaching thus becomes a performative act of identity construction ([63]). Framed by this view, the present study explores how multimodal classroom practices shape novice teachers’ evolving identities. [123] ([123]) account of language teaching methods further supports this inquiry by linking instructional choices to broader educational paradigms and teacher roles.

Despite a growing recognition of the importance of teacher identity and multimodal pedagogy, existing research predominantly focuses on experienced educators in Western contexts, leaving a critical gap in understanding how these strategies shape novice teachers’ professional identity development and their transition from novices to experienced practitioners ([89]). Specifically, there is limited empirical work on how Chinese novice university English teachers employ multimodal resources (e.g., gestures, head movements, eye gaze) to construct and adjust their professional identities in classroom interactions. By addressing this gap, this study examines how multimodal strategies facilitate their transition from authoritative to facilitative teaching roles. Guided by Norris’s Multimodal (Inter)action Analysis (MIA) framework ([102], [104], [107]), it explores the following research question: How do novice college English teachers’ teaching practices and professional identities evolve through multimodal interaction strategies? By analyzing the interplay between multimodal strategies and identity development within a specific cultural and institutional context, this study aims to deepen our understanding of novice teachers’ professional growth in diverse educational settings.

## 2. Literature Review

### 2.1. Teacher Identity and Pedagogical Practices: A Theoretical Overview

Teacher identity, broadly defined as “the way we make sense of ourselves and the image of ourselves that we present to others” ([33]), is integral to guiding teachers’ practices and professional growth ([36]; [147]). It is inextricably linked to pedagogical practices, serving as both a reflection of and a vehicle for professional growth ([58]; [125]). Language teacher identity has been a crucial factor in enhancing pedagogy and teacher learning ([72]; [84]), whose development is a dynamic, socially situated process influenced by factors such as institutional culture, peer collaboration, student interactions, and societal expectations, as well as professional interactions ([81]; [145]; [147]).

This intersection of identity and pedagogy has attracted increasing scholarly attention, particularly in understanding how teaching strategies reflect and reinforce evolving teacher identities ([80]; [149]). Initial strategies such as creating a safe learning environment, fostering strong student–teacher relationships, and using precise instructional methods are foundational in shaping teacher identities ([88]; [133]). These strategies provide a foundation for more complex approaches like adaptive teaching and integrating cultural awareness ([61]), emphasizing responsiveness to classroom dynamics and individual student needs ([57]; [119]). Moreover, teachers who are proficient in non-verbal communication (e.g., eye contact, gestures, body language) help establish stronger teacher–student rapport, enhance interactive idea exchange, and facilitate deeper comprehension ([117]; [137]; [146]).

The evolving teacher identity also directly impacts teaching quality and classroom dynamics ([28]). Scholars have found that a strong professional identity can enhance teachers’ confidence and commitment to their profession and decision-making, ultimately improving the quality of classroom teaching ([10]). For instance, teachers identified as “managers” may adopt more authoritative and structured teaching methods, potentially limiting student interaction ([60]). In contrast, teachers with a supportive, guiding identity often foster closer teacher–student relationships and more active engagement ([116]). Thus, teacher identity is critical in pre-service and in-service development, influencing curriculum design, guiding pedagogical enactment, and shaping classroom interactions ([46]). [49] ([49]) identify three overlapping identities among Business English educators—practitioners, researchers, and professionals—with each fostering distinctive pedagogical orientations. Similarly, curriculum orientation has been shown to shape identity development; practice-driven programs (e.g., in Hong Kong) promote stronger pedagogical agency than theory-heavy ones (e.g., in Guangdong) ([48]).

This review foregrounds the interplay between pedagogical agency and identity formation, particularly through embodied interaction, a lens further deepened through multimodal analysis in subsequent sections.

### 2.2. Multimodal Strategies as Sites of Identity Construction

The multimodal turn in language education has emphasized the need to move beyond speech and text to include gesture, gaze, image, posture, space, and other semiotic modes in meaning-making ([76]; [86]; [143]). Multimodal teaching has proven highly effective in language learning, fostering social competencies ([90]), critical thinking ([71]; [110]), intercultural awareness ([1]), and communicative engagement ([7]; [115]). Furthermore, some scholars focused on modality-specified features, such as language resources ([62]), facial expressions ([39]), vocal features ([127]), psychological reactions ([37]), or body language ([142]). Therefore, teachers’ multimodal competence is essential for effective teaching, especially when instruction differs from their native language ([97]). Research indicates that multimodal classroom interaction is driven by teachers’ pedagogical goals rather than their personal teaching style ([18]). By strategically using semiotic resources, teachers can create meaningful student experiences, enhancing teaching and learning ([40]). High levels of multimodal competence positively influence student engagement, particularly during classroom lead-ins ([121]), and multimedia-enriched environments improve comprehension and engagement ([120]). Yet, despite these developments, research has remained largely focused on student uptake and outcome. Studies rarely examine how teachers themselves negotiate identity through multimodal strategies in real-time classroom contexts.

Despite the documented benefits of multimodal pedagogy, its classroom implementation remains fraught with challenges. [163] ([163]) highlighted teacher reluctance, limited conceptual training, and philosophical resistance as key barriers, especially in EFL contexts. Similarly, [69] ([69]) noted that multimodal teaching in ESP settings often lacks theoretical grounding, places high cognitive and emotional demands on teachers and learners, and suffers from underdeveloped assessment frameworks for integrating semiotic resources such as gesture, gaze, and spatial layout. Such constraints underline the need for a deeper understanding of how multimodal practices mediate identity negotiation, particularly among novice teachers navigating complex educational transitions.

### 2.3. Negotiation of Teacher Identities: Novice English Educators in Chinese Classrooms

The professional journey of novice English teachers in China is shaped by both institutional structures and evolving educational philosophies. Traditionally, Chinese EFL classrooms have emphasized top–down, teacher-centered instruction, often following rigid Initiation–Response–Feedback (IRF) or Initiation–Response–0 (IR0) patterns ([139]). This identity in traditional, teacher-centered classrooms emphasizes teachers’ roles as lesson providers and class leaders, with students in the default role of passive learners ([124]). New teachers often adopt a “situated identity”, which refers to the way individuals construct their identities in specific social and institutional contexts ([169]).

However, with China’s educational reforms promoting student-centered approaches, novice educators face challenges in shifting from knowledge transmitters to facilitators of interactive, student-driven learning ([74]; [167]). EFL teachers are increasingly expected to focus on the teaching process and co-construct knowledge with students. As novice teachers progress, integrating intercultural competence (ICC) becomes vital for professional development and identity formation ([12]; [75]). As global citizenship becomes a curricular priority in China ([24]), English teachers increasingly embed intercultural reflection into their pedagogical practices ([23]), thereby repositioning themselves as cultural brokers who promote cross-cultural dialogue and global awareness ([32]; [118]). This transformative role is further reinforced through systematic intercultural training, which equips teachers with strategies to navigate culturally diverse classrooms and solidify their identities as mediators of intercultural learning ([54]). Ultimately, the shift from transmitting static linguistic knowledge to fostering dynamic cross-cultural understanding underscores the evolving identity of professional English educators in transnational contexts ([56]).

Yet, despite policy-level encouragement, the actual process through which novice teachers negotiate this professional transformation remains underexamined, especially in terms of their embodied classroom practices. The affordances of multimodal resources in helping novice teachers move from traditional to more responsive teaching identities warrant deeper empirical investigation—a gap that this study seeks to fill.

### 2.4. Analytical Framework: Multimodal (Inter)action Analysis

Given that professional identity is shaped through embodied, situated interaction, a methodology capable of attending to micro-level actions and contextual dynamics was essential. MIA was adopted for its capacity to integrate fine-grained analysis of communicative modes (e.g., gaze, gesture, posture, spoken discourse) with broader institutional and sociocultural structures ([107]). Grounded in discourse analysis ([20]), interactional sociolinguistics ([45]), mediated discourse analysis ([129]), and multimodality ([67]), MIA views all actions as identity-telling ([104]; [128]). It is particularly useful for examining how teachers construct identity through various modes.

At the core of MIA is mediated action, which sees social actors acting with or through cultural tools such as language, gestures, objects, or discourse ([107]). Mediated actions are categorized into lower-level actions (e.g., a gaze shift or utterance), higher-level actions (HLAs) composed of multiple lower-level actions, and frozen actions embedded in physical objects, such as notes on a blackboard ([102], [105]). For example, a teaching moment in an L2 class may involve a combination of gestures, gaze shifts, and written notes, with frozen actions preserving meaning in material form. Moreover, identity in this study is approached as situated and interactional, evolving within the specific institutional, cultural, and pedagogical structures of Chinese universities. MIA’s attention to both micro-level actions (e.g., pointing while asking a question) and higher-level aggregated practices (e.g., managing a discussion through gaze and gesture) enables a nuanced understanding of how novice teachers navigate between traditional teacher-centered expectations and emerging student-centered demands.

Methodological tools like scales of action, modal configuration, and modal aggregation are central to MIA. Scales of action reveal how smaller actions are embedded within larger ones, offering insights into how lower-level actions like asking questions or providing feedback support broader teaching objectives ([106]). Understanding these smaller actions within HLAs is essential for identifying effective teaching practices and improving instructional quality and effectiveness ([52]; [94]).

Figure 1 illustrates embedded actions relevant to class teaching, such as asking questions, asking students, and giving feedback. By examining how these actions interrelate, one can identify their immediate impacts and how they contribute to broader educational goals or shifts in constructing teachers’ professional identity.

Modal configuration refers to the dynamic arrangement of hierarchical or non-hierarchical modes used to produce higher-level actions ([102]). In teaching, this involves adapting verbal and non-verbal cues to different contexts. Multimodal inputs, integrating auditory, visual, and textual elements, enhance vocabulary retention more effectively than unimodal inputs ([96]). A key concept within this framework is modal aggregates, where multiple lower-level actions merge at the same hierarchical level to form a unified higher-level action ([108]). As the modal configuration is fluid, specific modes take precedence over others depending on context, affecting the dynamics of teaching actions and their role in facilitating learning ([103]).

As illustrated in Figure 2, modes of teaching evolve, with some gaining or losing prominence. The size of the circles represents their significance—more considerable for dominant actions, smaller for subordinate ones—while overlapping circles indicate simultaneous occurrence. For example, when a teacher speaks, gazes at students, and points at a PowerPoint slide, spoken language and gaze serve as primary actions, while pointing is secondary. This spoken language–gaze–point aggregate conveys a unified message, demonstrating how multimodal communication creates a combined effect greater than the sum of its parts ([65]).

When modes combine in a modal aggregate, they create a cohesive teaching message and reflect shifts in the use of MIA’s techniques over time. These shifts reveal changes in pedagogical strategies and identity development. The dynamic nature of modal configurations offers insights into how teachers adapt their multimodal skills. This aligns with the view of [6] ([6]) that multimodal texts result from the combined effects of all resources used to create and interpret them. Understanding modal configurations and aggregates is crucial for analyzing how teachers construct meaning, engage students, and shape their professional identities in classroom interactions.

Combined with an ethnographic sensibility, this framework enables a contextualized understanding of how identity is performed within and shaped by institutional settings ([16]). The research design followed an information-oriented case study logic ([42]), aiming for analytical richness rather than statistical generalization. The flexible, participant-responsive data construction process reflects the dialogic nature of classroom identity negotiation, which MIA is well-suited to capture through attention to real-time multimodal interaction ([31]).

A growing body of research confirms the suitability of MIA for analyzing classroom teaching. [2] ([2]) and [154] ([154]) demonstrated how students and teachers use multiple modes—gesture, gaze, posture, and voice—to make meaning in digital and science classrooms. [13] ([13]) focused on novice EFL teachers’ coordination of gaze and gesture, while [113] ([113]) emphasized that classroom discourse is always multimodal, shaped by embodied and affective dimensions. [21] ([21]) further showed that gaze synchronizes with speech and turn-taking, regulating classroom participation. Complementing these findings, [14] ([14]), [66] ([66]), [77] ([77]), and [78] ([78]) highlighted the pedagogical significance of spatial layout, image, and movement. [109] ([109]) extended MIA to case-based research, reinforcing its value for capturing identity formation and interaction in educational contexts. Together, these studies validate MIA as a robust framework for examining the multimodal nature of classroom pedagogy and teacher identity.

As mentioned above, these theoretical and methodological components position MIA as a uniquely suited framework for investigating the performative, dynamic, and semiotically rich nature of teacher identity in the classroom. It enables the tracing of how pedagogical practices, symbolic resources, and institutional discourses converge in the embodied enactment of professional roles.

## 3. Study Design

This study was approved by the Auckland University of Technology Ethics Committee (AUTEC Ref: 22/243). Written informed consent was obtained from all participating teachers. No students appeared in any visual recordings; classroom videos exclusively focused on teachers’ pedagogical behaviors. While incidental student voices were captured during interactions, these were neither analyzed nor processed (aligned with AUTEC’ s exemption for non-identifiable incidental data, Article 4.7). Institutional consent was secured from department heads prior to recruitment, and all procedures adhered to research ethics protocols for participant confidentiality and voluntary participation.

To address ethical concerns regarding power dynamics and participant vulnerability, this study was designed with reflexivity and protection in mind. To protect participant anonymity and avoid the risk of indirect identification, the two focal participants are referred to throughout the paper as Teacher A and Teacher B. After data analysis, participants were invited to member-check selected excerpts and interpretations. These post-study debriefings allowed them to clarify intent, verify accuracy, and ensure respectful representation.

Participant recruitment was conducted between 1 March and 19 July 2022, at a second-tier university in Anhui Province, China. Four early career English teachers were initially recruited based on the following three key criteria: (1) they were novice teachers within the first five years of their teaching careers; (2) they were voluntarily participating in the study; and (3) they were teaching College English curriculum at the research site. These criteria ensured consistency in institutional and pedagogical contexts while respecting participant agency. Such alignment enabled the meaningful comparison of identity trajectories across shared teaching practices.

For each participant, three 45-min lessons were recorded at the beginning (first recorded session), the middle (second recorded session), and the end of the semester (third recorded session), totaling 135 min per participant. Data analysis followed the systematic steps of MIA ([107]), including data delineation, visual transcription, micro-analysis, and the application of analytical tools to identify higher-level actions (HLAs) performed by the participants.

While all four teachers demonstrated varying degrees of pedagogical development, two participants (Teacher A and Teacher B) exhibited more substantial and sustained shifts in multimodal and student-centered practices. Their trajectories offered the richest insights into the study’s core aim: understanding how novice teachers develop multimodal pedagogies in EFL contexts. Thus, purposive sampling was employed to focus detailed analysis on these two cases. Initial analysis of all four participants ensured that these selected cases were representative and illustrative of broader patterns observed across the cohort.

Regarding generalizability, this study follows a qualitative tradition that prioritizes analytic generalization over statistical generalization ([159]). Rather than aiming for population-wide claims, it offers theoretically grounded insights that may inform future research and pedagogical training in comparable EFL contexts. The two focal cases provide depth over breadth, illustrating key processes in multimodal and identity development across a semester of classroom teaching.

Visual transcripts were constructed by extracting key moments from video stills and transcribing verbal and non-verbal modes (e.g., gaze, gestures, posture). This comprehensive approach provided insights into teaching strategies, classroom management, and intercultural activities, highlighting how modes contribute to meaning-making and identity construction. By using [107]’s ([107], [108]) transcription conventions and tools, such as waveforms and font size to represent intonation and loudness variations, the analysis revealed the interplay of discourses and practices within mediated actions and engagement sites.

To analyze multimodal interactions, it is crucial to understand how each mode contributes to meaning-making. A mode is “a system of mediated action with regularities” ([108]). Micro-analysis of modes—such as spoken language, layout, gesture, gaze, head movement, and facial expression—reveals how discourses and practices intersect within a site of engagement ([91]). These modes collectively shape identity construction and classroom communication. [107] ([107]) provides transcription conventions for spoken language, using commas for slight rising intonation, periods for lowering intonation, and dashes for glottal stops. Multimodal transcription incorporates tools like rhythmic pattern waveforms and font size for loudness variations ([107]). The following subsections explain how different modes can be analyzed:−*Pitch* is a mode of auditory modulation. Pitch enhancement can facilitate adult vocabulary learning across different visual contexts ([41]). By altering the pitch range, secondary school learners can be subtly guided without direct feedback ([132]). This study used Praat’s software (version 6.3.08) ([17]) to automatically generate pitch and intensity contours, enabling the precise analysis of acoustic properties such as pitch (frequency) and intensity (amplitude) over time ([17]).−*Layout:* “A mode that informs people about the distance between objects, the environment, and the people (inter)acting” ([107]). In a classroom, layout is reflected in how tables and chairs are arranged, along with elements like pictures, a blackboard, and a screen.−*Gesture*: “A mode that tells us how individuals hold and move their arms, hands, and fingers” ([107]). Teachers’ gestures in the classroom involve natural interactions with students or teaching tools, such as a blackboard, computer, or mouse ([80]).−*Gaze*: “A mode that tells us how individuals look at something or someone” ([107]). For example, when a teacher notices a student playing with their phone in class, he will look at the student to inform him not to do so.−*Head movement*: “A mode that tells us how individuals hold and move their heads” ([107]). For example, if the teacher wishes to explain the knowledge points in a PowerPoint, then they move their head from the students to the screen.−*Facial expression*: “A mode that tells us how individuals maintain and change their expressions on the face” ([107]). For example, when the teacher asks students questions, they always wear a smile to foster a sense of closeness.−*For data analysis part*, all data were transcribed and analyzed by the author following the MIA framework as outlined by [107] ([107]).

To enhance credibility and reduce subjective bias, two external researchers with training in multimodal interaction analysis independently reviewed 20% of the transcriptions and modal codings. Intercoder reliability (Cohen’s Kappa = 0.85) was calculated, ensuring interpretive reliability. Discrepancies were discussed until agreement was reached, ensuring intercoder reliability. Member checking was conducted with both focal participants after preliminary findings were produced, allowing them to comment on interpretations and adjust any misrepresentations of intent.

## 4. Findings

Drawing on sociocultural perspectives of teacher identity ([11]; [141]; [162]), this study identifies three evolving identity states—compliance, experimentation, and dialogic enactment—to capture the transformation observed in novice teachers’ classroom practices. These categories are empirically derived from repeated viewing and multimodal coding of classroom video data, rather than imposed as theoretical abstractions. The compliance phase involves textbook-centered instruction, minimal student interaction, and monologic delivery ([168]). Experimentation is marked by emerging, though inconsistent, use of praise, feedback, and student prompts—suggesting a negotiation between established routines and new pedagogical expectations ([27]; [138]). Dialogic enactment, by contrast, features the co-construction of meaning, sustained dialogic exchanges, spontaneous feedback, and embodied engagement through smiling, head nods, gaze, and posture shifts ([19]; [83]).

This three-phase development shows how teacher identity evolves through social interaction in specific classroom contexts. By tracing these identity shifts through fine-grained multimodal patterns, the study contributes to understanding how novice EFL teachers gradually move from performative instruction toward relationally and dialogically grounded teaching.

### 4.1. Shared Beginnings: Teacher-Centered Practices and Passive Knowledge Transmitters

In their first recorded sessions (Session 1), both teachers showed similar teacher-centered multimodal patterns. These practices show that both participants acted as passive knowledge transmitters. These behaviors aligned with the “chalk and talk” model, often seen as ineffective by students due to its monologic and disengaging nature ([140]).

In Figure 3, Teacher A asks, “In general, do people tend to have more confidence in themselves or others?” but answers them herself without inviting student responses. Her gaze was mostly directed at the textbook, with only brief eye contact (frames 1–7). Similarly, in Figure 4, Teacher B states, “You have finished listening to the text, and try to find the main idea of the message.” Without using nonverbal communication as a powerful tool to enhance interaction and knowledge transfer ([160]), Teacher B remains focused on the textbook, with minimal eye contact or body language directed toward the students (frames 2–9). Moreover, the heavy reliance on textual content with minimal student interaction reflects a lack of confidence in managing classroom dynamics ([112]; [126]).

In frames 8–11, Teacher A pauses for about a minute before posing the following rhetorical question: “It seems like it’s more towards others, right?” The direction of the arrow in frame 9 shows her continued focus on the textbook, and the timestamp indicates her lack of fluency during the lecture. Studies have shown that fluency in teaching positively affects student learning outcomes ([153]), with smoother speech patterns and fewer pauses enhancing vocabulary comprehension ([165]). Teacher A’s frequent pauses and reliance on the textbook suggest a lack of fluency, which could hinder students’ understanding and retention of the material. In frame 12, Teacher A’s self-adaptive gesture—brushing her hair in frame 12—suggests anxiety or self-soothing behavior typical of low teaching confidence ([59]; [101]).

In all excerpt figures, multimodal elements such as gaze (yellow arrows), gesture (red circles), head movement (red arrows), and posture (directional shifts) are annotated to highlight their temporal and spatial alignment with verbal discourse. This visualization strategy follows MIA conventions ([107]) to reveal shifts in identity performance.

These teacher-centered patterns reflect the use of grammar translation and audio-lingual approaches during Session 1, both of which prioritize controlled input and linguistic accuracy over communicative use. Notably, the primary instructional language was Chinese, which teachers used for vocabulary explanation, grammar clarification, and general instruction. English was used only intermittently—for isolated lexical items or reading aloud—rather than as a medium of sustained classroom interaction. This limited use of the target language may hinder students’ exposure to authentic input and reduce opportunities for language acquisition through use ([156]). The dominance of Chinese also reflects teachers’ early-stage identities as content deliverers, marked by a focus on correctness and authority rather than on fostering student engagement or fluency in English. Such practices are common in Chinese tertiary EFL contexts, where exam-oriented curricula and teacher authority norms often discourage the extensive use of English in class ([164]).

Overall, the teacher’s attention throughout this sequence is primarily on herself and the textbook rather than the students. This reflects a teaching approach characterized by instructional dominance, with a primary focus on content delivery rather than student participation.

### 4.2. Teacher A’s Identity Trajectory: From Compliance to Dialogic Enactment

At the outset, Teacher A exhibited a compliance-oriented identity, characterized by textbook-driven instruction, rhetorical questioning without student response, limited eye contact, and monologic delivery. By the end of the semester, Teacher A demonstrated a clear shift toward dialogic enactment, guiding her students to critically reflect on their learning and encouraging them to co-construct rather than passively learn the knowledge ([155]). This change reflects the broader pedagogical shift towards critical thinking and collaborative learning, where teachers are no longer just transmitters of knowledge but partners in the learning process ([5]).

In Figure 5 (frames 1–5), Teacher A asks, “In the context of cross-cultural interactions, when facing cultural conflict, what attitude should we adopt?” Teacher A’s question goes beyond traditional language-focused topics such as vocabulary, sentence structure, and translation. Instead, it reflects a deeper understanding of intercultural awareness, which is not merely about recognizing cultural differences but requires critical reflection on one’s biases and the ability to flexibly adapt in cross-cultural settings ([34]). By asking students to consider appropriate attitudes, Teacher A encourages them to analyze the underlying values, beliefs, and social norms that shape cultural conflicts, fostering intercultural competence. Her question aligns with the model of Intercultural Communicative Competence ([23]), which emphasizes the importance of self-awareness and the ability to shift perspectives in intercultural interactions. By encouraging students to consider underlying values and social norms, she fosters higher-order thinking through analysis, synthesis, and evaluation ([151]). Such questioning techniques are crucial for promoting critical and creative thinking, a central goal of culturally responsive pedagogy ([99]). Her multimodal orchestration becomes more evident in frames 7–18. She calls on multiple students sequentially, using smiling, sustained gaze, and encouraging intonation ([53]; [68]; [131]).

In frames 19 and 20, Teacher A repeats the student’s explanation while looking up, a practice known as “revoicing” ([26]). This repetition acknowledges the student’s contribution, validates their response, and allows the entire class to engage with and expand on the idea, ultimately transforming it into shared knowledge ([38]). Following this, in frames 22–24, she offers explanation feedback by not simply judging correctness but deepening understanding, a strategy known to enhance long-term conceptual learning ([22]).

### 4.3. Shifts in Scales of Actions and Modal Configurations: A Comparative Analysis of Teacher A

Figure 6 compares the scales of actions in Teacher A’s first and third recorded sessions. In the first recorded session, Teacher A’s HLAs during the lecture consisted of two primary actions, asking a question and answering the question. In contrast, in the third recorded session, her actions expanded to include asking a question, waiting for an answer, asking a student, and summarizing answers.

Figure 7 illustrates the comparison of modal configurations in these two sessions. In the first recorded session, when Teacher A asks and answers questions by herself, object handling, gaze at the textbook, and spoken language function together. In contrast, gaze at students does not overlap with other modes. This pattern reflects a traditional, teacher-centered approach, where information is transmitted one way with limited engagement ([30]). However, the third recorded session demonstrates a notable shift in modal configurations. Teacher A’s spoken language, facial expressions, gaze toward students, and head movements became more pronounced when addressing students, particularly during discussions. This transformation aligns with student-centered learning principles, emphasizing active participation and improved learning outcomes ([15]).

Each circle represents a communicative mode, with its size indicating the degree of prominence in each teaching moment—larger circles denote dominant modes, smaller ones represent supporting modes. Overlapping circles signify simultaneous use. For instance, when a teacher speaks, maintains eye contact, and points to a slide, spoken language and gaze may dominate, while gesture supports, forming a cohesive multimodal ensemble.

The following figures present the spectrograms of Teacher A’s first session (Figure 8 and third session (Figure 9), illustrating the development of her sound wave, pitch, and intensity while she delivers lectures.

The comparison between Teacher A’s first and third recorded sessions highlights a significant evolution in her vocal delivery patterns, as shown by Praat-generated spectrograms and acoustic analysis ([17]). In the first session (Figure 8), Teacher A’s speech was restrained and formally neutral, characterized by a narrow pitch range (180–220 Hz) and stable intensity levels (±2 dB), reflecting a teacher-centered lecturing style focused on clarity and authority. By the third session (Figure 9), her vocal patterns shifted notably during the question “What attitude should we adopt?”; her pitch range expanded (150–300 Hz), and intensity variation increased (±6 dB), indicating heightened emotional investment and a more conversational tone. This enhanced vocal expressiveness aligns with a student-centered approach, fostering interactive discussions and critical thinking through dynamic vocal engagement ([51]). The widened pitch contour and intensity peaks (Figure 9) further illustrate her transition from knowledge transmission to active learner involvement.

### 4.4. Teacher B’s Identity Development: Navigating Experimentation to Enacted Dialogism

Teacher B’s professional growth similarly unfolded across distinct identity configurations. Initially, her instructional identity aligned with compliance, marked by monologic delivery, fixed gaze on the textbook, and minimal interpersonal engagement. As her confidence developed, she entered a phase of experimentation, where she tentatively incorporated student praise, evaluative feedback, and personal storytelling. In the final phase, Teacher B achieved dialogic enactment by fostering reciprocal interactions between students and teachers ([43]; [85]).

This transformation is exemplified in Figure 10. In frames 1–6, Teacher B acknowledges student responses through a relaxed body posture, direct gaze, and affirming language, such as “Very good.” These verbal and nonverbal cues, including head tilting, eye contact, and a calm demeanor, not only reinforce student participation but also model approachability and support ([4]). In frames 7–8, Teacher B adds praise, “very constructive suggestions”, while sweeping her gaze across the room to engage the whole class. Such explicit evaluative language, like “very constructive suggestions,” effectively reinforces student contributions and encourages broader participation ([87]). In frames 13–18, she shares an anecdote about a previous student majoring in data science, highlighting his success with social media posts that have garnered hundreds of thousands of likes. Sharing such personal stories helps bridge the gap between teachers and students, making the classroom environment more relatable and fostering engagement ([70]; [130]).

As the semester progressed, both teachers began to experiment with more student-centered practices, reflecting a gradual shift in both pedagogical orientation and professional identity. These changes were closely associated with the adoption of communicative language teaching (CLT) ([25]) and task-based language teaching (TBLT) strategies ([144]). Specifically, we observed increased use of open-ended questions, more frequent opportunities for peer discussion, and the incorporation of real-world topics into classroom tasks. Importantly, these interactions were conducted primarily in English, marking a notable departure from the Chinese-dominant instruction observed in Session 1. This linguistic shift not only increased students’ exposure to authentic language input but also signaled the teachers’ growing confidence in facilitating learning through target language use. Such practices align with what [79] ([79]) describes as “communicative-oriented competence” and reflect a move away from transmission toward interaction as a core instructional mode.

### 4.5. Shifts in Scales of Actions and Modal Configurations: A Comparative Analysis of Teacher B

Figure 11 compares the scales of actions in Teacher B’s first and third recorded sessions. In the first session, Teacher B’s HLAs primarily involve asking a question. In the third session, the actions expand to include not only asking questions but also giving feedback and sharing stories.

Figure 12 illustrates the modal configurations in Teacher B’s HLAs across sessions. In the first session, Teacher B adopted a teacher-centered approach, relying heavily on the textbook, with object handling, gaze at the textbook, and spoken language functioning together, leading to limited student interaction. By the third session, she shifted to a student-centered approach, integrating gaze on the students, facial expressions, spoken language, and head movement into a cohesive modal aggregate, fostering direct interaction. This progression underscores Teacher B’s professional growth, highlighting the value of interactive techniques for early career teachers to enhance teaching effectiveness and cultivate an adaptive teaching identity.

The following figures present the spectrograms of Teacher B’s first session (Figure 13) and third session (Figure 14), illustrating the development of her sound wave, pitch, and intensity while she delivers lectures.

By comparing Teacher B’s voice intensity and pitch between the first and third sessions, notable changes can be observed. In the first session, the teacher’s tone remains relatively flat with minimal variation in both pitch and intensity. However, in the third session, there is a marked increase in pitch, particularly when Teacher B says, “Very good, very constructive suggestions.” This shift in pitch and intensity variations suggest a more dynamic and engaged teaching style.

By comparing Teacher B’s vocal patterns between the first and third sessions, distinct shifts in delivery emerge. During the initial session, her speech maintains a restrained quality, marked by a narrow pitch range and steady intensity levels, consistent with a formal, teacher-centered approach. By the third session, however, her vocal dynamics shift notably, particularly during interactive exchanges such as her response, “This insight really pushes our discussion forward.” Here, her pitch rises sharply (e.g., expanding from a baseline of 100 Hz to 250 Hz) alongside pronounced intensity fluctuations (±8 dB), reflecting heightened expressiveness and emotional investment. The comparison highlights the evolution of Teacher B’s teaching approach over the semester, possibly reflecting greater enthusiasm and adaptability in response to student interaction ([50]).

As shown in Table 1, both Teacher A and Teacher B exhibited increased multimodal complexity and interactivity in Session 3. Notably, student-directed gaze, use of gestures, and open questions all rose significantly, indicating a shift toward dialogic, student-centered teaching.

## 5. Discussion

This study introduces a novel three-phase model of professional identity transformation—compliance, experimentation, and dialogic enactment—derived from longitudinal classroom observations of novice EFL teachers. This framework offers a new lens for understanding how teachers negotiate institutional expectations and pedagogical reform through multimodal strategies, including gaze, gesture, spatial positioning, and voice modulation. By applying MIA to capture higher-level actions, action scales, and modal configurations, the study demonstrates how novice teachers orchestrate these semiotic resources to reposition themselves from traditional knowledge transmitters to facilitators of collaborative meaning-making.

First, this study contributes to theoretical understandings of novice teacher identity by demonstrating how multimodal classroom practices mediate professional growth. Whereas much of the identity literature conceptualizes identity as narratively constructed or discursively negotiated ([147]; [150]), this study advances the field by illustrating how identity is materially enacted through the fine-grained orchestration of communicative modes. Unlike most studies employing MIA, which primarily focus on students’ learning processes ([93]; [152]), this research uniquely centers on novice teachers’ professional identity development. For instance, Teacher A’s transition from a passive knowledge transmitter towards a facilitator identity, exemplified by her focus on intercultural discussions, underscores the importance of guiding students beyond content mastery towards developing global awareness ([8]). Likewise, Teacher B’s development of a story-sharing identity was enacted not through formal instruction, but via personal anecdotes, embodied enthusiasm, and positive evaluative feedback ([100]). These findings expand symbolic competence models ([64]) by showing how multimodal conduct becomes central to enacting and transforming professional identities.

Second, this study offers practical insights for enhancing EFL teacher education, particularly in non-Western, exam-driven contexts. Building on [98]’s ([98]) findings, which emphasize the importance of raising teachers’ awareness of multimodal classroom interaction and fostering systematic reflection on teaching practices, our data further suggest that novice teachers benefit most when they are explicitly supported in developing both multimodal sensitivity and reflective routines. While [35] ([35]) and [148] ([148]) underscore the value of multimodal strategies, they seldom explore how these strategies transform teacher self-positioning in real-time. This research addresses that gap by showing how modal frequency, pitch variation, and multimodal alignment reflect and facilitate shifts in pedagogical stance. To support such transformation, teacher education programs should cultivate novice teachers’ awareness of how gaze direction can regulate interaction ([131]), gestures can scaffold abstract content ([134]), and vocal modulation can enhance student attention and classroom affect ([132]). Although many studies have identified the early years of teaching as a period marked by identity confusion ([55]), pedagogical insecurity ([92]), and even regression ([135]), the present study demonstrates how multimodal strategies can instead foster confidence, promote pedagogical agency, and enable early-stage positive identity development. In contrast to reflective models grounded in verbal or written self-reporting ([111]; [122]), this study proposes that conscious attention to multimodal conduct offers an alternative route for identity development, especially in cultural contexts where explicit self-disclosure is less normalized.

Third, this study makes a methodological contribution by operationalizing MIA in longitudinal classroom settings. While many teacher identity studies rely on interviews, reflective journals, or thematic discourse analysis ([9]; [161]), this research demonstrates how visual transcripts, pitch spectrograms, and fine-grained multimodal coding reveal transformation processes that are otherwise difficult to capture. Furthermore, unlike many multimodal studies that focus on student learning ([73]; [136]), this study centers teachers’ multimodal agency and demonstrates that novice teachers reposition themselves—often unconsciously—through micro-interactional adjustments. The findings thus affirm and extend MIA’s proposition that identity is formed through mediated action and semiotic layering over time.

## 6. Conclusions

Through MIA, this study has illuminated how two novice university EFL teachers in mainland China navigated evolving identity configurations—compliance, experimentation, and dialogic enactment—as they embraced increasingly interactive and student-centered pedagogical practices. Situated within an institutional culture heavily influenced by examination-oriented traditions, both participants initially adopted authoritative, textbook-driven roles that positioned them as knowledge transmitters rather than facilitators. As the semester progressed, however, sustained engagement with multimodal strategies such as sustained gaze, vocal expressiveness, embodied feedback, and interactive questioning enabled them to reconfigure their teaching identities, moving toward more relational, interculturally responsive, and pedagogically adaptive roles.

While this study focuses on two novice EFL teachers at a single Chinese university, the findings may be transferable to other exam-driven and resource-constrained EFL contexts, such as rural Chinese institutions or Southeast Asian universities. The identity shifts observed—from compliance to dialogic enactment—reflect challenges common to novice teachers navigating pedagogical reforms. However, transferability is shaped by local cultural and institutional conditions, including teaching traditions and assessment pressures. As such, while the analytical framework and identity phases proposed may inform teacher education elsewhere, adaptation to context is essential.

However, several limitations must be acknowledged. First, the relatively small sample size and single institutional setting restrict the generalizability of the findings. Second, this investigation predominantly emphasizes teachers’ practices without incorporating student voices or broader institutional factors that undoubtedly influence identity transformation. Third, the timeframe was limited to a single semester, providing only a temporal snapshot of an inherently dynamic and prolonged developmental trajectory.

Future research should address these limitations by investigating a broader array of institutional contexts, incorporating multiple stakeholder perspectives, especially those of students and institutional policymakers, and by adopting longitudinal methodologies to capture identity transformations more holistically. Exploring how novice teachers negotiate their identities over extended periods and across diverse educational settings would further enrich our theoretical and practical understanding of teacher development.

## Figures and Tables

**Figure 1 behavsci-15-01050-f001:**
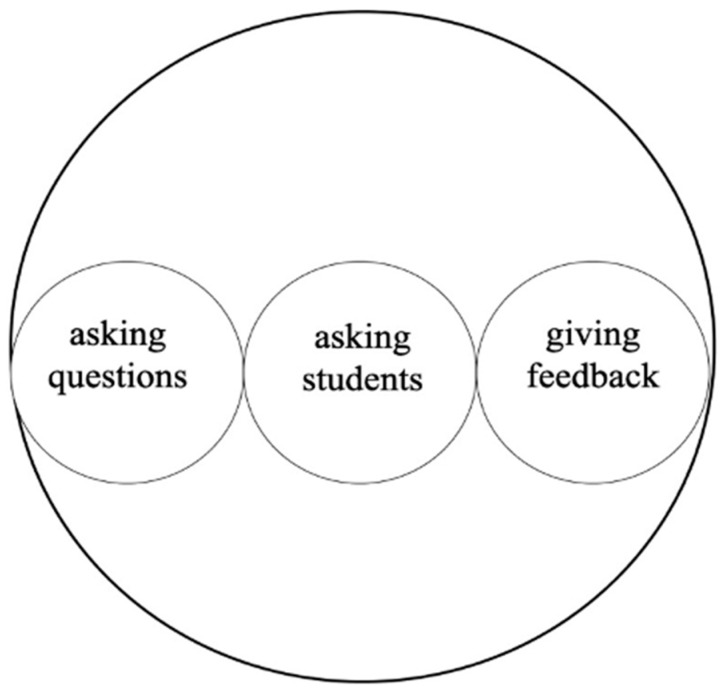
Example of embedded actions in in-class teaching.

**Figure 2 behavsci-15-01050-f002:**
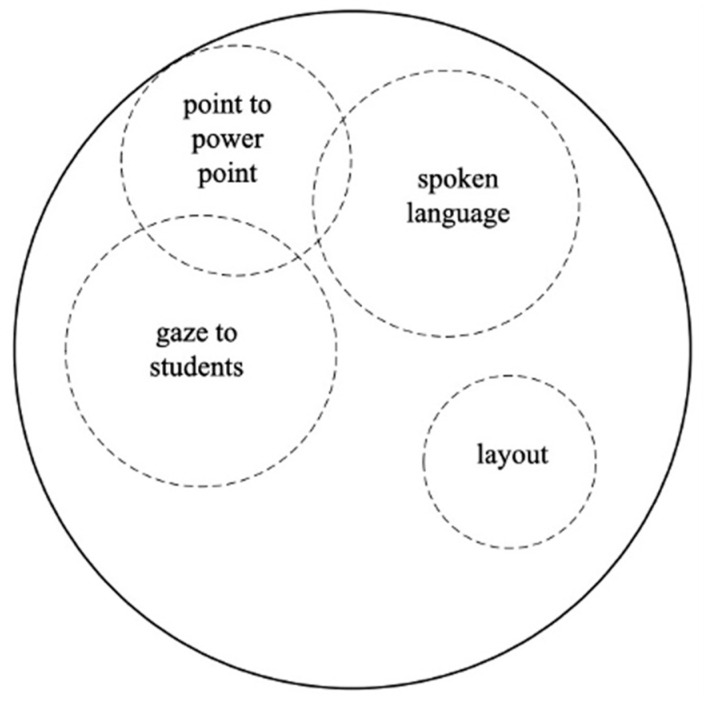
Example of modal configuration and aggregate of HLAs in word-explaining.

**Figure 3 behavsci-15-01050-f003:**
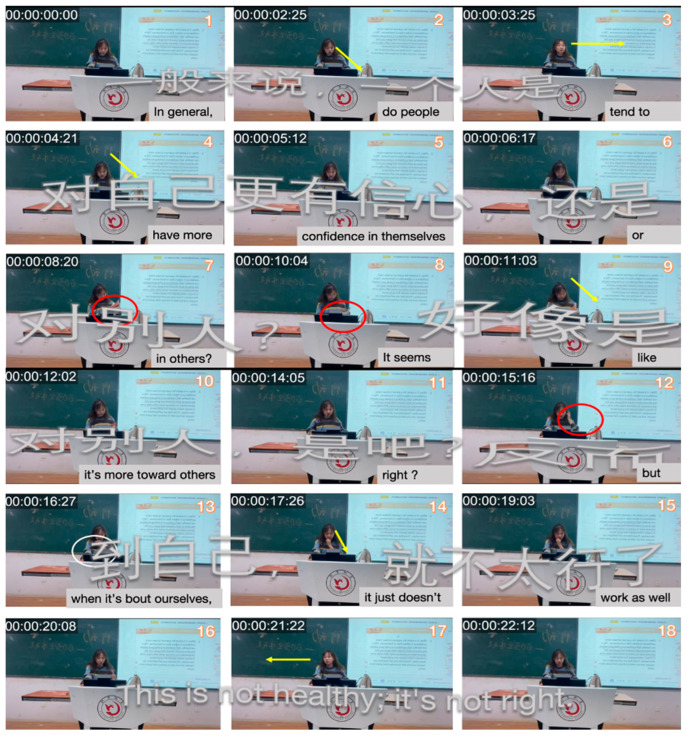
Excerpts of Teacher A’s teacher-centered modality in Session 1.

**Figure 4 behavsci-15-01050-f004:**
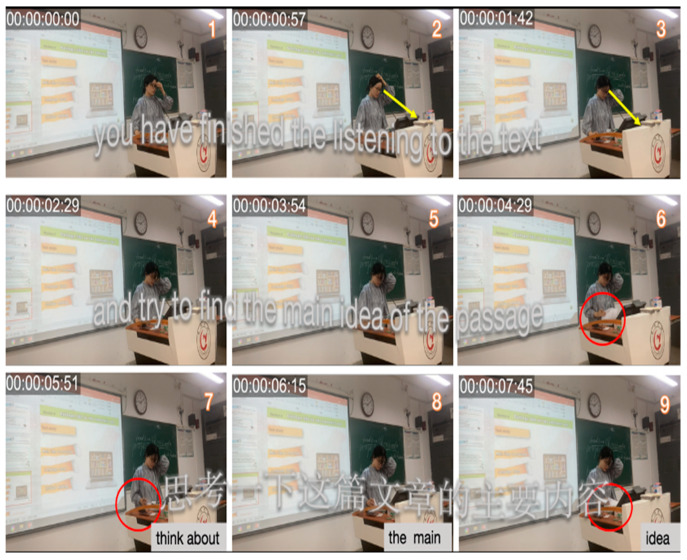
Excerpts of Teacher B’s teacher-centered modality in Session 1.

**Figure 5 behavsci-15-01050-f005:**
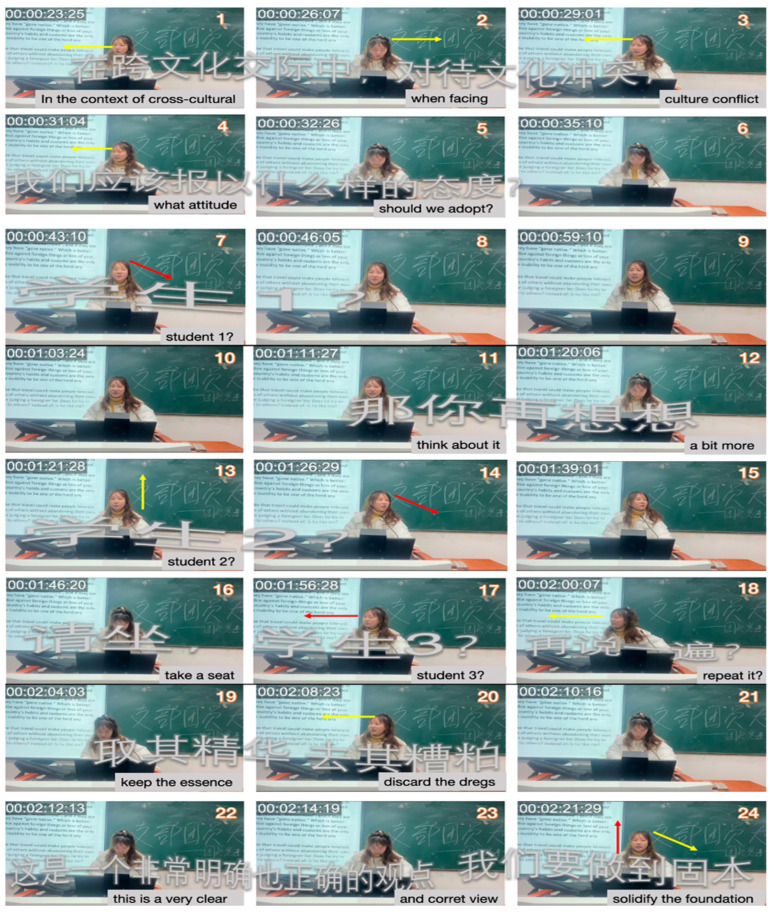
Excerpts of Teacher A’s facilitation of intercultural discussion in Session 3.

**Figure 6 behavsci-15-01050-f006:**
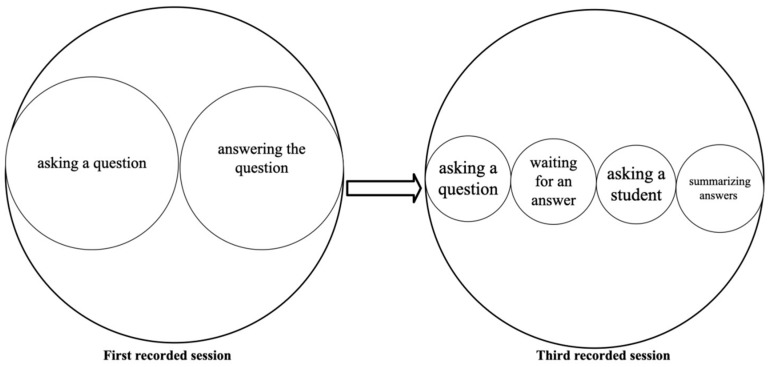
From monologic to dialogic: changes in Teacher A’s scales of actions.

**Figure 7 behavsci-15-01050-f007:**
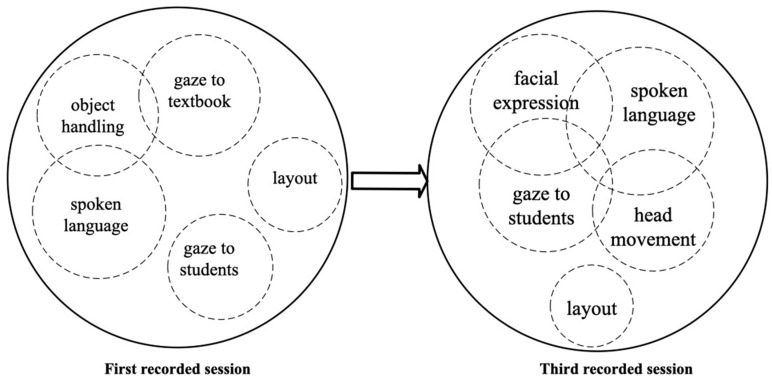
From isolated modes to connected modes: progression in Teacher A’s modal configuration.

**Figure 8 behavsci-15-01050-f008:**
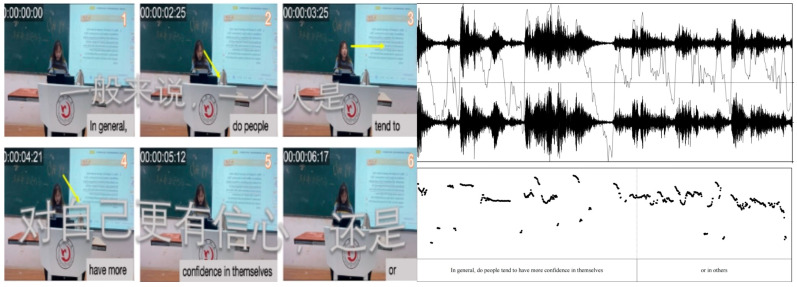
Flat pitch and intensity in Teacher A’s Session 1.

**Figure 9 behavsci-15-01050-f009:**
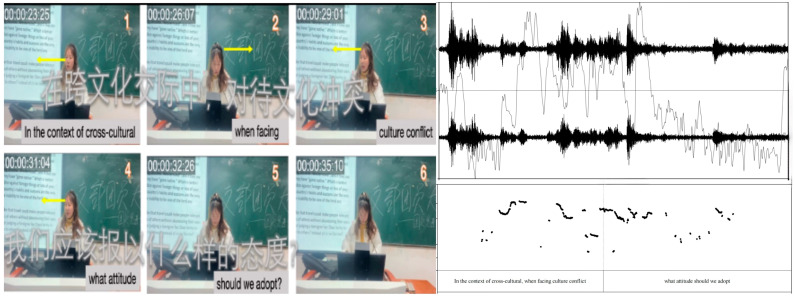
Wider pitch and expressive range in Teacher A’s Session 3.

**Figure 10 behavsci-15-01050-f010:**
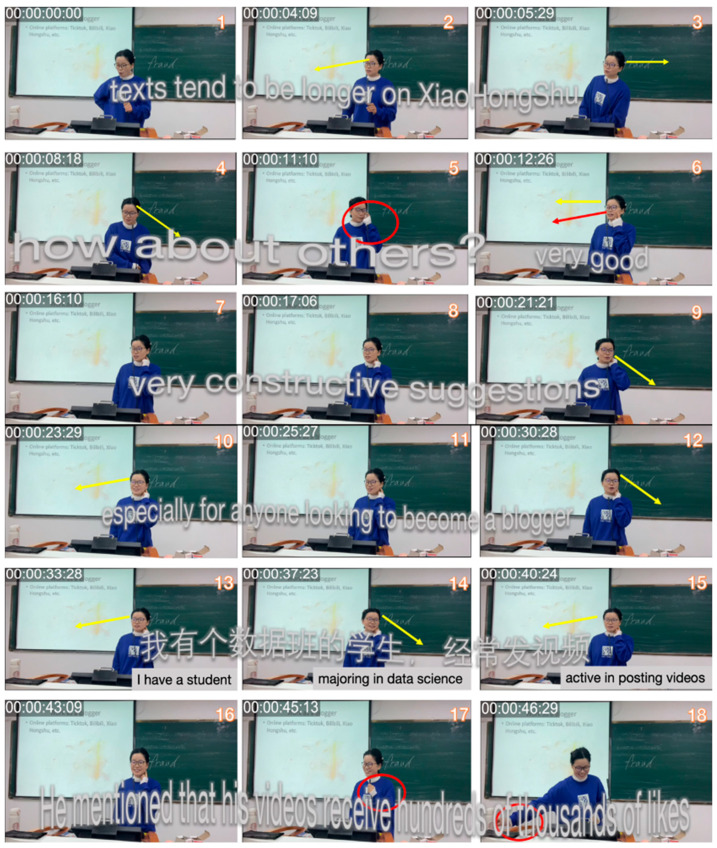
Excerpts of Teacher B’s increased interaction and multimodal responsiveness in Session 3.

**Figure 11 behavsci-15-01050-f011:**
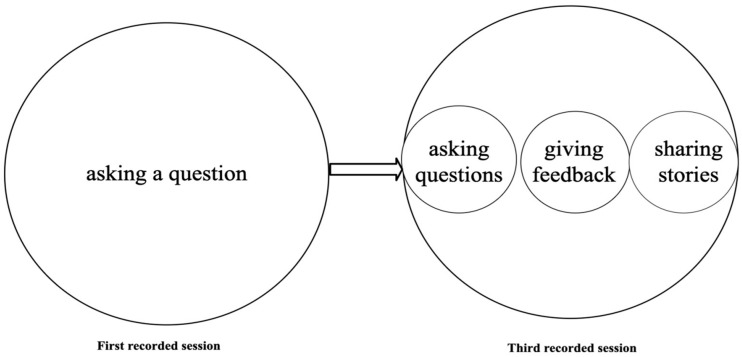
From questioning to connection: changes in Teacher B’s scales of actions.

**Figure 12 behavsci-15-01050-f012:**
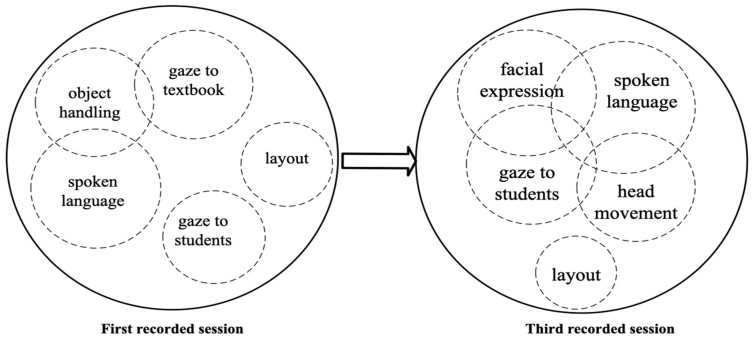
From isolated to coordinated interaction in Teacher B’s modal configuration.

**Figure 13 behavsci-15-01050-f013:**
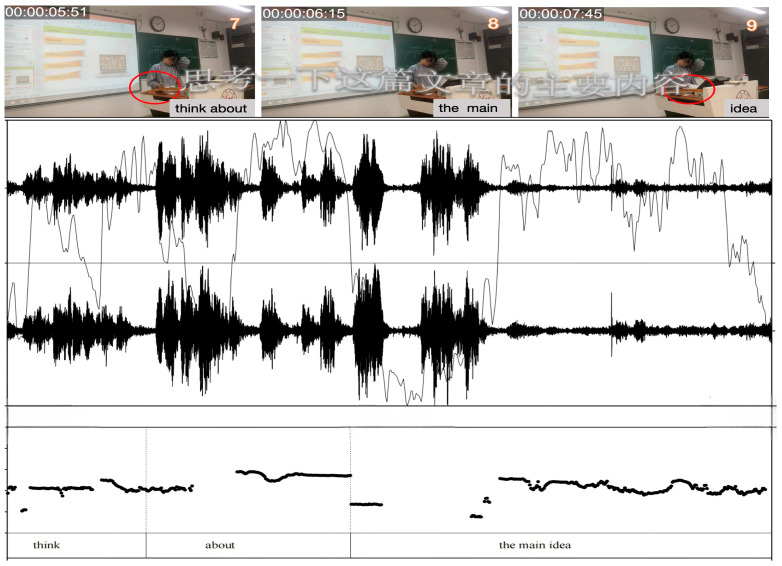
Restrained vocal delivery in Teacher B’s Session 1.

**Figure 14 behavsci-15-01050-f014:**
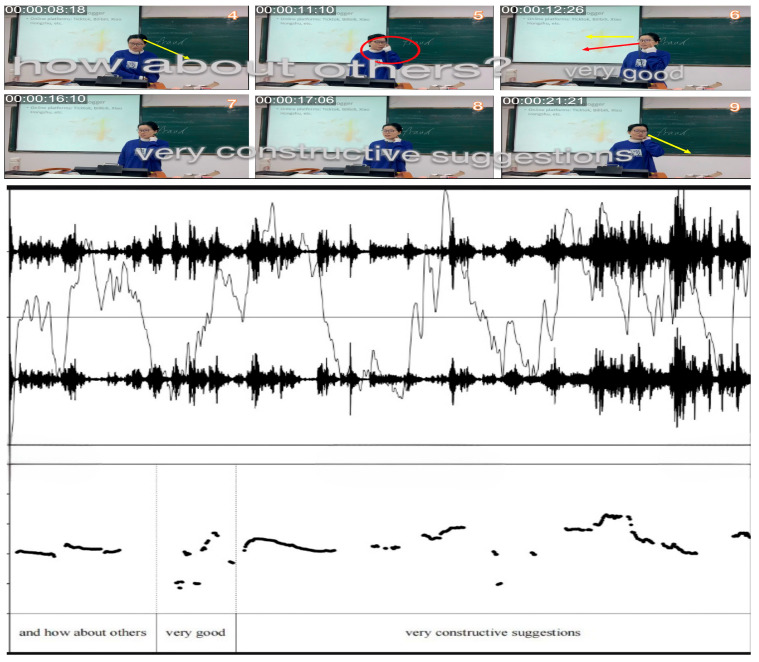
Dynamic and expressive delivery in Teacher B’s Session 3.

**Table 1 behavsci-15-01050-t001:** Comparison of modal frequencies in Session 1 and Session 3 (based on 10 min excerpts from each session).

Total Modal Shifts	Teacher A S1	Teacher A S3	Teacher BS1	Teacher BS3
Teachers’ talk time	9 min (mostly lecture)	6 min (more dialogue)	6 min (3 min playing listening recording)	5 min (more peer discussion)
Student-directed gaze	2	9	2	11
Gesture (self-adaptors excluded)	3	10	2	12
Feedback moves	0	3	1	6
Open question	0	3	0	3
Students’ response	0	3	0	5

## Data Availability

The datasets generated and analyzed during this study are part of the author’s PhD research and are currently unavailable, as the PhD is ongoing. Data may be available from the corresponding author upon reasonable request after completing the PhD.

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
