# Peer review of "Transforming Pedagogical Practices and Teacher Identity Through Multimodal (Inter)action Analysis: A Case Study of Novice EFL Teachers in China"

_behavsci, 2025, doi:10.3390/bs15081050_

Round 1
Reviewer 1 Report
Comments and Suggestions for Authors
Dear Authors,
Thank you for your interesting and clearly written manuscript. The topic—exploring novice teachers’ identities through multimodal classroom analysis—is promising and relevant. However, several key areas require further development in order to strengthen the article's scholarly contribution.
Most notably, the manuscript would benefit from a stronger theoretical engagement with the field of foreign language teaching in higher education. Scholars such as Claire Kramsch (2014) offer critical insights on the intersection of language, identity, and culture that could deepen the analysis.
Additionally, the use of a multimodal model focused on gaze and gestures, while innovative, may not fully capture the complexity of teacher identity—particularly for foreign language teachers. The paper also lacks contextual detail about language teaching in Chinese higher education, and the pedagogical framework (e.g., references to “lesson delivery”) seems outdated.
Finally, the identity typologies introduced at the end (traditional vs. student-centered) need better theoretical grounding and a clearer link to foreign language pedagogy.
I encourage you to :
- Deepen the literature review to engage with key scholarship in foreign language pedagogy and teacher identity (e.g., Kramsch, Richards & Rodgers).
- Clarify the appropriateness of the multimodal model for identity research, and justify its use in the context of foreign language teaching.
- Provide more detailed contextual background on English language instruction in Chinese higher education.
- Reflect on and revise the use of pedagogical terminology to align with current educational paradigms.
- Strengthen the theoretical framing of teacher identities and clarify their relevance to foreign language education.
- Specify the methods and approaches used in the observed classrooms, including language teaching methodology and actual use of the target language.
Reviewer 2 Report
Comments and Suggestions for Authors
The topic is interesting and relevant. Below, I share some comments that I believe could help improve the paper:
- Although is mentioned that four teachers initially participated and only two were analyzed in depth, this decision is not clearly explained.
- The introduction states that there is a lack of research on teacher identity and multimodality among novice teachers, but no concrete evidence to support this claim.
- The Background and Findings repeat concepts already introduced or re-explain previously (multimodal modes or shifts in teaching roles..), making it necessary to reduce redundancies and reorganize the content.
- The manuscript does not include limitations or future research directions.
- The suggestions for teacher training programs are general and should be further specified.
- It would be advisable to state the contribution of the study more explicitly. At present, it's implicit in the discussion.
Reviewer 3 Report
Comments and Suggestions for Authors
Please see the attached document

Round 2
Reviewer 2 Report
Comments and Suggestions for Authors
There is a noticeable overall improvement in the writing, an expanded theoretical framework, and the inclusion of limitations and future research directions. The reference list has also been enriched, and some pedagogical implications have been more articulated.
However, I would like to make observations that could still be refined:
It would be advisable to state the specific contribution of the study more clearly. Currently, it remains implicit in the discussion and is not sufficiently clearly distinguished from previous research.
I suggest ensuring consistency in the term Multimodal Interaction Analysis (MIA). Once the full term is introduced, only the acronym MIA should be used in the rest of the manuscript. On line 250 the full term appears again, you have to revise all document.
Best regards,
Author Response
Comment 1:
“It would be advisable to state the specific contribution of the study more clearly. Currently, it remains implicit in the discussion and is not sufficiently clearly distinguished from previous research.”
Response 1:
Thank you for this insightful and constructive comment. In the revised manuscript, we have significantly restructured and expanded the Discussion section to ensure that the specific contributions of the study are clearly and explicitly articulated, and to better distinguish our findings from previous research.
In particular, we made the following key revisions (p19-20,line 598-655):
-
Theoretical Contribution Clarified (Paragraph 1):
We now explicitly state that the study advances current theoretical understandings by illustrating how identity is not only discursively or narratively constructed (Varghese et al., 2005; Wang et al., 2024), but is also materially enacted through the orchestration of communicative modes. We emphasize that, unlike most MIA studies which focus on student learning (Mejía-Laguna, 2023; Wigham & Satar, 2024), this study centers on novice teachers’ professional identity development, which represents a novel analytical focus. -
Practical Contribution Expanded (Paragraph 2):
We incorporated findings from Morell (2020) and further elaborated on how novice teachers benefit from explicit support in developing both multimodal sensitivity and reflective routines. We also highlighted how our study goes beyond the general endorsement of multimodal strategies (as seen in de Oliveira & Barbosa, 2023; Veliz et al., 2024), by providing detailed empirical evidence of how such strategies transform teacher self-positioning in real-time. -
Methodological Contribution Highlighted (Paragraph 3):
We clarified that the study offers a methodological advancement by operationalizing MIA in a longitudinal teacher identity study—using visual transcripts, pitch spectrograms, and fine-grained multimodal coding. We contrasted this approach with traditional reliance on interviews or thematic discourse analysis (e.g., Barkhuizen, 2016; Yuan & Mak, 2018), and explained how our data reveal transformational processes that might otherwise remain invisible. We also emphasized that the study addresses the underexplored area of teachers’ multimodal agency, distinct from the dominant focus on students’ multimodal engagement in previous work (Li, 2020; Tan & Matsuda, 2020).
We hope these revisions clearly address your concern and make the contributions of the study more transparent, distinctive, and impactful.
Comment 2:
“I suggest ensuring consistency in the term Multimodal Interaction Analysis (MIA). Once the full term is introduced, only the acronym MIA should be used in the rest of the manuscript. On line 250 the full term appears again, you have to revise all document.”
Response:
Thank you for pointing this out. We have carefully reviewed the entire manuscript and revised all instances to ensure consistent use of the acronym MIA after its first introduction. The full term Multimodal Interaction Analysis no longer appears after its initial definition, including the instance noted on line 250.
Reviewer 3 Report
Comments and Suggestions for Authors
The revised manuscript demonstrates substantial improvement and now presents a compelling, theoretically grounded, and well-documented case study. The progression from teacher-centered to dialogic teaching is convincingly illustrated. The authors' use of MIA, including modal configurations and pitch analysis, enriches the discussion.
Author Response
comment 1:The revised manuscript demonstrates substantial improvement and now presents a compelling, theoretically grounded, and well-documented case study. The progression from teacher-centered to dialogic teaching is convincingly illustrated. The authors' use of MIA, including modal configurations and pitch analysis, enriches the discussion.
response 1: We sincerely thank Reviewer 3 for the thoughtful and encouraging feedback. We truly appreciate your recognition of the manuscript's theoretical grounding, the clarity of the teaching progression from teacher-centered to dialogic modes, and the contribution of our MIA-based analysis. Your comments were both reassuring and motivating, and they have reinforced our confidence in the value of the study.
Thank you once again for your time and support.